# Demo: Knowledge Graph-Based Housing Market Analysis

**Abstract.** The housing market is complex and multi-faceted, which makes its analysis challenging for users and professionals. We develop a four-step knowledge graph-based knowledge extraction approach for the housing market for efficient and accurate data analysis, consisting of data acquisition and cleaning, entity linking, ontology mapping, and question answering. The proposed system allows one to summarize the housing information for a selected geographical area, analyze the surroundings by collecting census data, understand the medical safety based on COVID-19 data, and the area attractiveness based on celebrities data from DBpedia. Our system can provide personalized recommendations given keywords and information about the market over time. A user-based evaluation demonstrates the utility of our system.

**Keywords:** Knowledge graph, DBpedia, COVID-19, Recommendation

## 1 Introduction

Analyzing the housing market involves data collection from various websites, narrowing down search based on geographical or economical constraints, and iterative searching over the collected information. This process is highly time-consuming and laborious, and becomes infeasible when the data size is large. In this paper, we develop a knowledge graph (KG)-based tool for flexible analysis of the housing market, given user constraints. Specifically: C1 We develop a KG pipeline for integrating information about housing items and their nearby environment, including economic data, COVID-19 data, nearby celebrities, and transits, etc. C2 We develop a demo-tool given the geography of interest, which uses three representative methods, based on: keywords, entity resolution and rule-based matching, and Latent Dirichlet Analysis. The tool supports flexible user searches: fixed search, queries, and a combination of the two. C3 We demonstrate the utility of the tool with two use-cases: rental recommendations based on user queries and analysis of the spatio-temporal dynamics of the market.

The KG technology plays a central role in this application. The data enrichment, either through adding new data from other sources or further extraction from the original sources, is native to KGs [3]. As an integrated data source, it enables easy analysis and recommendation algorithms to be developed. Existing KG visualization functionality helps us to study correlations in the market and predictive market analysis.

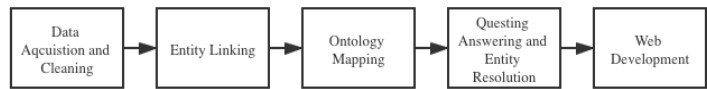

**Fig. 1.** Housing KG Construction Pipeline.

## 2 Proposed Knowledge Integration Approach

The goal of our system is to produce an integrated knowledge graph. Our architecture is shown in Fig. 1. First, data acquisition and data cleaning prepare usable and abundant data. Then data from different sources is linked based on similarity, allowing one to explore implicit relationships between the sources. After defining appropriate ontology nodes, relationships, classes, and properties, we map the extracted knowledge to this ontology. Once the knowledge graph is ready, we build a question answering module to support intuitive queries over our graph. Finally, all the data interactions are integrated into a user-friendly web application, developed by full-stack technology.

Without loss of generality, we implement the above pipeline for the use case of apartment finding in the state of California to consider a geographical constraint. We describe the design and implementation of the pipeline.

### 2.1 Data Acquisition and Cleaning

Four sources of data are leveraged in our project: 1) over 24k records of apartment information from Apartments[1] and ApartmentFinder[2], including properties like location, floor plan, phone, and rating; 2) more than 55k records of celebrities' information, such as names, birthplaces, death places, hometowns, residences, alma maters, from DBpedia [1]; over 3k records of US census data and COVID-19 data from the Census Bureau[3] and Los Angeles Times[4], including properties such as race distribution, average family income, employment rate, the number of confirmed cases of COVID-19, and the number of deaths.

We circumvent crawling challenges like anti-crawler, dynamic web, and the limits on the number of returned records by setting sleep times, using different user-agents, and generating dynamic IPs. We perform further normalization of the original data. The duplicates and missing values are processed by dropping records or filling them with the average or median. The cleaned data are used for further data integration.

### 2.2 Entity Linking

Entity linking is the process of establishing identities between representations in different sources, which is critical to the semantic integration of sources. It is

---

[1] https://www.apartments.com/

[2] https://www.apartmentfinder.com/

[3] https://www.census.gov/

[4] https://www.latimes.com/projects/california-coronavirus-cases-tracking-outbreak/

challenging due to name variations and ambiguity [5]. The same apartment may have different names and address formats on different websites, so similarity calculation is needed to link the records which represent the same entity. We compute Levenshtein similarity[7] of the apartment name and location: $Sim = \alpha * SimLocation + (1 - \alpha) * SimTitle$. In our work, we set $\alpha$ to 0.9, giving higher weight for location as the records with the same location are always identical. We set the candidate threshold to 0.8, which means if $Sim$ for two entities is greater than 0.8, then they are counted as candidates. Then, the highest similarity record is marked as linked. We use RLTK[6] to implement the similarity functions[5].

Since the calculation is time-consuming with millions of pairs to be compared, blocking is used to reduce complexity [2]. We use zip code to divide apartments into corresponding blocks to only compare the apartments within the same county. Thus, the time consumed per county is reduced ten times.

The Us census data, Covid-19 data and celebrities data are linked with apartments by using zip code.

### 2.3  Ontology Mapping

Ontology mapping is defined as a process to find semantic correspondences between similar elements of different ontologies [4]. The semantic model of our KG is shown in Fig. 2. We have 10 classes with 46,200 instances. The main classes are: `ApartmentFinder_Info`, `Apartments_Info`, `University`, `Celebrity`, and `Location_Info`. There are 160,824 edges belonging to 13 types. COVID-19 and US census data

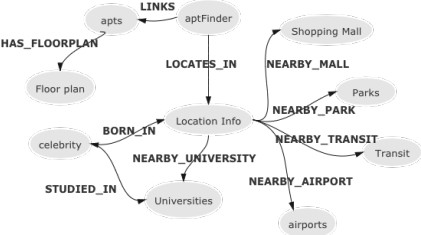

**Fig. 2.** The Semantic Model of our KG.

such as confirmed cases, confirmed death cases, crime rates, married ratio and population ratio are set as properties for location information. We store our graph in Neo4j.[6]

### 2.4  Query Matching

Three methods were used to process user queries: keywords, entity recognition/rule-based matching, and topic modeling.

**Keywords**: We attempt to map user keywords to defined categories, as shown in Table. 1. Some queries can be directly run on the KG, e.g., if shopping is mentioned, we order the apartments by the number of nearby shopping malls. Further indicators are defined by ourselves, e.g., for places with high security, we assume the confirmed cases, confirmed death cases of COVID-19, and crime rates should all be below the median.

---

[5] https://rltk.readthedocs.io/
[6] http://neo4j.org/

**Table 1.** Examples for Categories, with their Keywords and Indicators.

| Category | Example of Keyword | Indicator |
|---|---|---|
| education | university, education,.. | high school degree(%) & bachelor degree(%); number of nearby universities |
| security | safe, crime,security,... | confirmed case & deaths of Covid-19; crime rate |
| economy | rich, economy,.. | employ rate; mean income of families |
| age of apartment | new, built, time,... | built in time of apartment |
| social | friend, young,... | median age; the proportion of married |
| race distribution | Asian, black,... | the proportion of Asian/black/white |
| transportation | transit, traffic,... | the number of nearby transits |

**ER and Rule-based Matching**: We use pattern-based extraction to find the address, location name, and person name mentioned in the user query. The patterns include the surrounding words, whether they contain numbers, upper case, 'GPE' or 'LOC' entity type, etc.

**Topic Modeling**: If the query consists of paragraphs of description, we try to detect its topic, and return apartments which belong to the topic. To achieve it, we used a property named apartment.describe as raw data. Then NLTK[7] is applied to do data preprocessing, such as tokenization, removing stop words, lemmatizing, stemming, and filtering out words that appear infrequently. We used Gensim[8] to train LDA models with bag of words and TF-IDF. We evaluated the models and fine-tuned parameters based on the average scores for the most-matched topics. For instance, for the data in Los Angeles, we divided the apartments into 20 topics and the highest average score is 68.5%. Some of the keywords of topics are "Hollywood", "furnished", "park", etc.

## 3 Analyses

We show the utility of the proposed system with three use cases: intuitive visualization of housing information, analysis between housing aspects over time, and personalized housing recommendations.

**Visualization of Apartment and Surrounding Information** An intuitive view of an apartment and its neighborhood can be easily visualized based on user queries. As shown in Fig. 3, the URLs of apartments are connected with nearby shopping malls, parks, universities, the location (zipcode) node and so on. And the celebrities are linked with location information, while all the COVID-19 data and US census data are set as properties for the zipcode node.

**Factor Comparison over Time** Analysis about how data changes over time and the correlation between them can be easily performed. The proposed system can be used to track and analyze data changes between different counties, states, or countries. And the convenience of adding new data allows us to

---

[7] https://www.nltk.org/

[8] https://pypi.org/project/gensim/

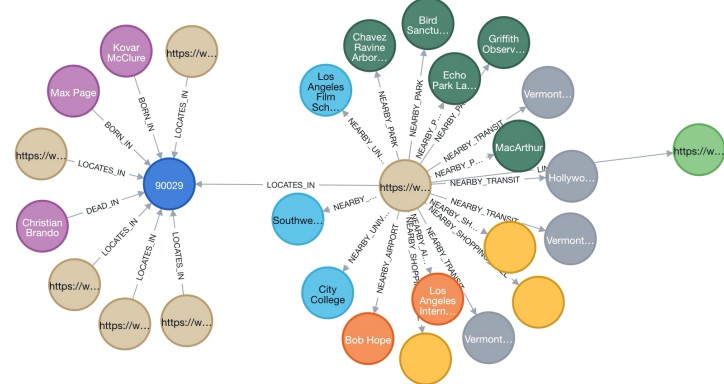

**Fig. 3.** Visualization of a Relevant Subgraph for a Query.

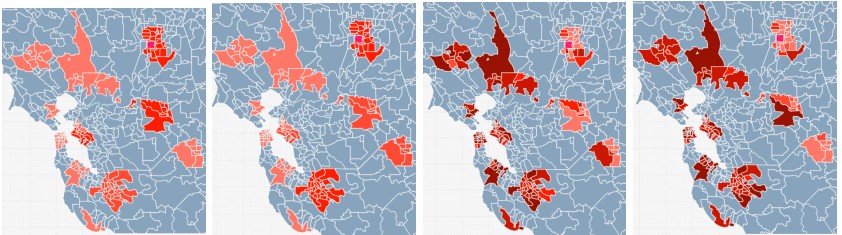

(a) Deaths in Nov.  (b) Deaths in Feb.  (c) Price in Nov.  (d) Price in Feb.

**Fig. 4.** Heatmap of Deaths of Covid-19 and Average House Price Changes.

continuously expand data over time. These may contribute to analyzing some issues related to marketing or social science, including factors that impact housing prices, the impact of COVID-19, and changes in the living standards, etc.

Fig. 4 compares the changes in the number of deaths of COVID-19 and the average house price per district between November 2020 and February 2021, for different counties in California. Darker color means a higher proportion of deaths, or higher average house price.

**Apartment Recommendation System** We also provide a recommendation interface to search for rental properties[9]. The user interface supports three exploration ways for users: 1) fixed search, e.g., "an apartment with *3 bedrooms*, less than *3500 dollars*"; 2) input queries, e.g., "a *modern designed* apartment near *6201 Hollywood Blvd* or near *Bernard cooper*, with *high security*. I can go for a walk in the nearby *park* on weekends. And the people here are *highly educated* and *relatively rich*"; and 3) a combination of both. The specific information about the apartments/houses that meet the requirements and nearby areas will be returned according to their score.

---

[9] https://github.com/ZepeiZhao/KG_APPLICATION

For evaluation, we designed a user evaluation aimed at the above three ways. Nine scenarios assigned for each user who was asked for making their own input on our website based on the scenarios. Then we compared the top 50 results from 10 users with standard results and calculated precision and recall.

The precision of fixed Search, queries, and their combination, are 1.0, 0.43, and 0.71, respectively. And the recalls of them are 1.0, 0.92, and 0.97, respectively. It is expected that fixed search provides more accurate results while information extracted from the query is less stable. The evaluation result proves that our implementation is useful.

## 4  Conclusion

In this paper, we developed a knowledge graph-based housing analysis pipeline and tool by: data crawling, data cleaning, entity linking, ontology mapping, and question answering. The advantages of building KG include systematic data integration, reuse, and personalized recommendation given input queries. The use cases that we provide demonstrate that our system can be used for a wide range of housing market analysis tasks. We expect that our approach could easily be reused for novel uses cases, like analyzing the impact of regional education and climate on China's housing prices, by merely adapting the data sources.

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
