# OpenReview forum: "Demo: Knowledge Graph-Based Housing Market Analysis"
_eswc-conferences.org/ESWC/2021/Workshop/KGCW — KGCW 2021_

### Official Review · ~Mario_Scrocca1 · 2021-04-13
**Complete demo of a knowledge graph for housing market analysis**

**Rating:** 7
**Confidence:** 4

**Review:**

The demo paper describes a complete pipeline demonstrating the construction and exploitation of a knowledge graph for housing market analysis. The discussed demo is relevant for the workshop and describes a comprehensive implementation: (i) the pipeline to build the graph involves data crawling from different data sources, data cleaning and entity linking (ii) the different data sources are then mapped to a unique schema in a graph database and a question answering/entity resolution mechanism is implemented on the populated knowledge graph (iii) the paper also describes a limited evaluation of a recommendation service built on top of the knowledge graph. I think the demo can also foster the discussion during the workshop since it integrates heterogeneous data sources (from DBPedia and other websites) in a knowledge graph supported by a label property graph.

There are however some issues that I recommend to address in the final version in case the paper is accepted:
- Section 2.3 doesn’t explain how the mapping process has been implemented and how the data sources have been integrated in Neo4j. Moreover, I would avoid using the term “ontology mapping” because, as I understand it, the process is more related to define a mapping between heterogeneous data sources and a unique graph schema.
- Related to the previous comment, I think that in the introduction of Section 2 the following sentence should be rephrased: “After defining appropriate ontology nodes, relationships, classes, and properties”. The terms listed overlap and, in general throughout the paper, I wouldn’t refer to the graph schema defined as an ontology.
- The link in the paper points to an empty repository. I assume this is the right link (from the same user) https://github.com/ZepeiZhao/APT_renting_platform_Knowledge_Graph
- The preliminary user evaluation is described very "hastily". My suggestion is to add at least a few sentences to clarify an example of the “scenarios” proposed to the users and how the authors selected the “standard results” used to compute precision and recall.

Minor issues
- List of the authors is missing before the abstract
- Add a bullet list or use italic for C1, C2, C3 to improve the readability of the different contributions in the introduction
- Fig. 1 I would remove “Construction” in the caption given that the last two steps represented are not related to the construction of the KG
- It is not clear how the main classes listed in Section 2.3 map to the labels in Fig. 2. E.g., I assume “apts” is a label for “Apartments_Info” but a note to clarify this should be added
- Typos
    - Fig 1. “Data aqcuisition” -> “Data acquisition”
    - Fig 1. “Questing Answering” -> “Question Answering”
    - Some spaces are missing before the references (e.g. [7] and [6])

---

### Official Review · ~Mohamed_Nadjib_Mami2 · 2021-04-14
**Demo paper describing a housing market analysis methodology that bases on many state-of-the-art algorithms but that lacks focus on the novelties or the most relevant components**

**Rating:** 8
**Confidence:** 4

**Review:**

The paper presents housing market analysis methodology based on several state-of-the-art algorithms and using several different data sources. Leveraging Knowledge Graph principles for data integration, this paper is of high relevance to the workshop.

I'd like to share with the authors some remarks:

- The paper goes too deep into answering the "what" and significantly neglects the "how". While it is understandably difficult to achieve such a balance in a short paper format, the significantly high number of methodologies, algorithms, steps and components leave the author with many questions. As examples, how mappings to the general ontology are defined and used; what are the types and formats of the original data sources; among existing graph representations, why Neo4j/property-graph is used., etc. Ultimately, even demo papers should still not look like a listing of popular algorithms and technologies, but rather elaborate on novelties and achieve a certain degree of focus.
- Regarding the evaluation, the above concern holds; we would appreciate seeing more details, e.g. the time performance of the different pipeline stages, the size of the data in terms of disk footprint, etc. Further, market analysis is a topic that has the potential of scaling very fast, so a metric of scalability is worth consideration.
- URLs "URLs of apartments" should rather be URIs, since not all entities are expected to have a representative corresponding web locator/page.
- Vague terms and expressions should be avoided or otherwise elaborated, e.g., "without loss of generality", "data cleaning prepare usable and *abundant* data".
- "describe as raw data" appears to be an unintended copy-paste into the text.
- There is no mention as to how the solution stands in comparison to similar efforts.
- Check the consistency of letter cases, e.g. "Us census data, Covid-19 data" ⇒ US Census, COVID-19.
- Finally, reusability, availability and reproducibility are very minorly touched upon, which would otherwise immediately improve the value of the paper, especially that you appear to make your source code publicly available (although the Github link you shared points to the wrong repo).

Overall, and ultimately, less breadth and more depth in the most relevant or significant components would have been more welcome.  Nevertheless, the paper is well written and presented describing a solid work that stands on the shoulders of giants; it also makes a great case of the use of KG technology in extracting value from multiple data sources.

---

### Official Review · ~Ben_De_Meester1 · 2021-04-19
**Too high-level description of a complete KG pipeline that is not self-standing enough to understand its impact**

**Rating:** 6
**Confidence:** 4

**Review:**

This demo paper describes a KG generation and usage pipeline for the housing market.
It is unclear what the demo during the workshop will contain:
the high-level architecture and system is described,
however, no technical details are given,
and the referenced repository https://github.com/ZepeiZhao/KG_APPLICATION does not contain any further information.
Given it's a demo paper, I would have expected a description of what the demo during the workshop will contain: will it be showcasing the Neo4J graph, some advanced queries you can do with the KG, demonstrating some use cases based on the web application, all of the above?
No clarification or argumentation is given for the choices of components or technology,
nor is there any link to an online demo, screencast,
or evaluation data.
The overall systems sounds interesting and useful,
however, the paper itself contains too little proof or reproducibility to fully understand the impact of the contribution, in my opinion.

EDIT: some more searching makes me assume that you probably wanted to link to https://github.com/ZepeiZhao/APT_renting_platform_Knowledge_Graph, which also links to a screencast video. This greatly improves the reproducibility of your paper 😅. A clear license description would be appreciated though, together with some more installation instructions.

Having gone through that repository, it does become more clear that more detailed descriptions of certain components are lacking. For example, given it's the Knowledge Graph Construction workshop, I'm very interested in how you actually generate the knowledge graph.
Based on your repository, it seems that all data processing/cleaning/linking happens mostly as preprocessing steps, which results in CSV files (details I would like to have seen in the paper).
The question I have is: how do you go from the CSV files to the actual graph? Is that via manually loading it in a Neo4J graph and manually mapping the data?
Is this documented somewhere? (I didn't find a description of the ontology/data model)
I also assume you're going for labeled property graphs instead of RDF, what's the reason? Specifically given one of your data sets is already an RDF knowledge graph instead of a LPG 🙂.

---

### Meta-Review · Program_Chairs · 2021-04-20

**Recommendation:** Accept
**Confidence:** 5

**Metareview:**

Dear authors,

Your work presents an interesting use case of knowledge graphs and is a valuable contribution to our community. Here are some improvements possible to the paper. It would nice if they are included in the camera-ready version of the paper.

- https://github.com/ZepeiZhao/APT_renting_platform_Knowledge_Graph seems to be the incorrect link.
- How the knowledge graph is generated is not explained in detail.

---

### Decision · Program_Chairs · 2021-04-23

Accept